


# Brief communication: Forecasting extreme precipitation from atmospheric rivers in New Zealand

Daniel G. Kingston[1], Liam Cooper[1], David A. Lavers[2,3], David M. Hannah[3,4]

[1]School of Geography, University of Otago, Dunedin, 9054, New Zealand
[2]European Centre for Medium-Range Weather Forecasts, Shinfield Park, Reading, RG2 9AX, UK
[3]School of Geography, Earth and Environmental Sciences, University of Birmingham, Birmingham, B15 2TT, UK
[4]Birmingham Institute for Sustainability & Climate Action, University of Birmingham, Birmingham, B15 2TT, UK

*Correspondence to*: Daniel G. Kingston (daniel.kingston@otago.ac.nz)

**Abstract.** With mountainous topography and exposure to mid-latitude westerly storms causing frequent atmospheric river landfall and associated hydro-hazards, medium-range forecasting of extreme precipitation is a critical imperative for New Zealand. Here, the European Centre for Medium-Range Weather Forecasts Extreme Forecast Index (EFI) is applied to two variables in forecast week 2, total precipitation (TP-EFI) and vertically integrated water vapour transport (IVT-EFI). Results reveal the TP-EFI sometimes outperforms the IVT-EFI in capturing extreme precipitation events – in contrast to past Europe-
based research and indicating a need to develop further our conceptual understanding of the predictability of extreme precipitation in different geographical contexts.

## 1 Introduction

Located in the mid-latitudes and with a relatively narrow east-west extent, New Zealand is strongly influenced by maritime air masses and embedded frontal weather systems. Combined with orographic forcing from steep topographic gradients, these
weather systems can result in the occurrence of substantial precipitation events. Indeed, with mean annual precipitation >12,000 mm in places (Kerr et al., 2018), the windward slopes of the Southern Alps on the South Island of New Zealand can be an exceptionally wet landscape.

Although New Zealand is impacted typically by the remnants of three to five tropical cyclones each year (Sinclair, 2002), most extreme precipitation and river flow events are associated with atmospheric rivers (ARs; Prince et al., 2021; Reid
et al., 2021; Kingston et al., 2022). ARs are narrow corridors of relatively intense atmospheric water vapour transport that are associated with extreme precipitation in the mid-latitudes globally, with New Zealand a hotspot for their occurrence (Guan and Waliser, 2015).

The precipitation that is associated with weak-to-moderate AR events is mostly beneficial (e.g. Ralph et al., 2019); in New Zealand these ARs are important for seasonal snow and, in turn, winter tourism and water resources for hydroelectricity
and irrigation schemes (e.g. Jobst et al., 2022; Porhemmat et al., 2021). However, more extreme AR events and their associated





precipitation can result in major disruption, with examples in recent years ranging from damage to major transport infrastructure (e.g. NIWA, 2019) to large-scale flooding of urban residential areas (e.g. NIWA 2021a). Correspondingly, accurate medium-range forecasting of such events – defined as a 3–14-day lead time – has clear benefits in terms of societal preparedness (including impact prevention, protection, mitigation, response and recovery).

35         Previous research (focussed mostly on western Europe) has found that in some cases the forecasting of extreme precipitation events associated with ARs can be more successful when the forecast focus is on the vertically integrated horizontal water vapour transport (integrated vapour transport, IVT) rather than precipitation itself (e.g. Lavers et al., 2016, 2017, 2018). Notably, this was found for the late medium-range forecast horizon (i.e. forecast week 2). This higher forecast skill has been associated with the more predictable large-scale nature of IVT in comparison to precipitation, and the strong

connection between IVT and precipitation in mid-latitude regions where precipitation occurs primarily at frontal boundaries. Interestingly, Lavers et al. (2016) found that the extent to which IVT-based forecasts outperformed those based on precipitation was influenced by characteristics of the large-scale atmospheric circulation (in particular, the North Atlantic Oscillation, NAO). For instance, week 2 IVT-based forecasts were most useful during the positive phase of the NAO – a difference attributed to the associated windier and stormier conditions and thus greater importance of IVT for precipitation.

45         Given these potentially valuable results in a western European context, it is important to determine the potential of IVT to improved extreme precipitation forecasts in other mid-latitude regions. New Zealand makes for a particularly interesting study region, given the dominance of frontal rain within the midlatitude westerlies, its status as a global AR hotspot, and the peculiar situation of ARs landfalling not only on prevailing windward (i.e. western) coastlines, but also northern and eastern coasts. Correspondingly, this brief communication explores – for the first time – the potential for IVT-based forecasts to

improve medium-range warning of extreme precipitation events in New Zealand according to synoptic situation and AR landfall location – focusing on three recent and highly damaging AR-related events.

## 2 Data and Methods

To provide focus for this Brief Communication, case study events are presented for three of the most damaging recent AR-related extreme precipitation events over New Zealand. Each event was associated with a markedly different synoptic-scale

meteorological situation. These correspond broadly to the dominant AR landfalling sectors documented for New Zealand by Prince et al. (2021). Event 1 was centred on the South Island east coast and included record magnitude 24- and 48-hour rainfall totals at a number of weather stations, peaking above 500 mm in 48 hours. The event exceeded a 200-year return period in places. Event 2 occurred on the South Island west coast, with the Buller River and coastal town of Westport particularly affected. Rainfall in Westport exceeded its monthly average in the space of 48 hours (212mm cf. 139 mm), and higher rainfall

totals in the upper Buller catchment resulted in exceptionally high river flows: at 7640 $m^3s^{-1}$ this marked the highest discharge ever directly recorded in New Zealand (NIWA, 2021b). The third case study event was focussed in the north of the South



Island. This was a longer duration event, with daily rainfall totals in places exceeding 100 mm for each of four consecutive days resulting in widespread flooding, particularly in the city of Nelson.

The IVT situation associated with each case study event was determined using ECMWF ERA5 IVT data, with additional information on the atmospheric circulation situation via the 850 hPa geopotential height field. The ERA5 reanalysis (Hersbach et al., 2020) provides a comprehensive record of the global atmosphere, land surface, and ocean waves on a 31 km (TL639) horizontal grid. For further context, mean sea-level analyses from the Meteorological Service of New Zealand (MetService) were also obtained.

This analysis focusses primarily on the European Centre for Medium-Range Weather Forecasts (ECMWF) Extreme Forecast Index (EFI; LaLaurette, 2003; Zsoter, 2006; Zsoter et al., 2015) that was used previously by Lavers et al. (2016, 2017, 2018). The EFI compares the probability distribution of forecasts with that of the model climate, thus highlighting regions that are forecast to experience anomalous weather. EFI values range from −1 to 1, with −1 implying extremely low and 1 implying extremely high values with respect to the model climate. EFI values > 0.5 indicate anomalous weather (Cox and Lavers 2020), but this threshold is lower at longer lead times because EFI values tend to diminish with increasing time into the forecast

horizon due to increasing forecast uncertainty (or spread). Herein, the EFI for IVT and total precipitation (TP) is considered for the ECMWF ensemble forecasts at two lead times. Our focus is on forecast week 2, based on previous findings (e.g., Lavers et al., 2016) that the IVT-EFI begins to outperform the TP-EFI for forecasting extreme precipitation at this time horizon. Specifically, our focus is the aggregated timescales of 7-9 days and 10-15 days, as the periods for which EFI data are archived by ECMWF. Although a more precise understanding of EFI differences may be possible from analysing daily time steps, these

data are not currently calculated operationally for the EFI. At this initial stage, analysis focussed on comparing the severity of EFI values and coherency of the spatial patterns in the EFI for TP and IVT.

## 3 Results

### 3.1 Easterly Airflow (Event 1)

During event 1 (South Island east coast) an easterly onshore wind and moderate IVT values were present, as part of a low-

pressure system moving from northwest to southeast (Fig. 1a). IVT at landfall on the east coast peaked at approximately 500 kg m$^{-1}$ s$^{-1}$, and 800-900 kg m$^{-1}$ s$^{-1}$ further upwind. This magnitude of IVT corresponds to a moderate AR in the scale of Ralph et al. (2019). Although length and width dimensions of high IVT values meet those typically used for identification of an AR (e.g. Guan and Waliser, 2015; Ralph et al., 2018), the area of high IVT follows a more cyclonic trajectory compared to the archetypal windward/west coast AR in North America or western Europe. It is this cyclonic trajectory that enables landfall on

the leeward (east) coast in a region of prevailing westerlies.

    Both IVT- and TP-EFI forecasts identify the possibility of an extreme event over the mid-to upper-South Island east coast at the 7–9-day period. The area of extreme IVT-EFI values is confined to the coast and region immediately offshore; whereas, the area of extreme TP-EFI values extends much further eastwards (Fig. 2, upper row), corresponding more closely





to the pattern of high IVT values (Fig. 1a) and a concurrent occluded front. As indicated in the difference plot, TP-EFI values
are up to 0.4 higher across the entire east coastal region.

At the 10–15-day period there is little evidence of a forthcoming extreme event (Fig. 3, upper row). There is a region
of IVT- and TP-EFI values in the 0.2-0.4 range, and although lower EFI values are expected at longer lead times due to
increasing forecast uncertainty, this is still below the 0.5 threshold used typically to signify an extreme event. Additionally,
for the IVT-EFI these values are located further north than the actual event. While TP-EFI values are higher (by 0.1-0.2) over
the case study region, they are still <0.5 and their location is not well-matched to the eventual spatial pattern of raised IVT
values or extreme precipitation.

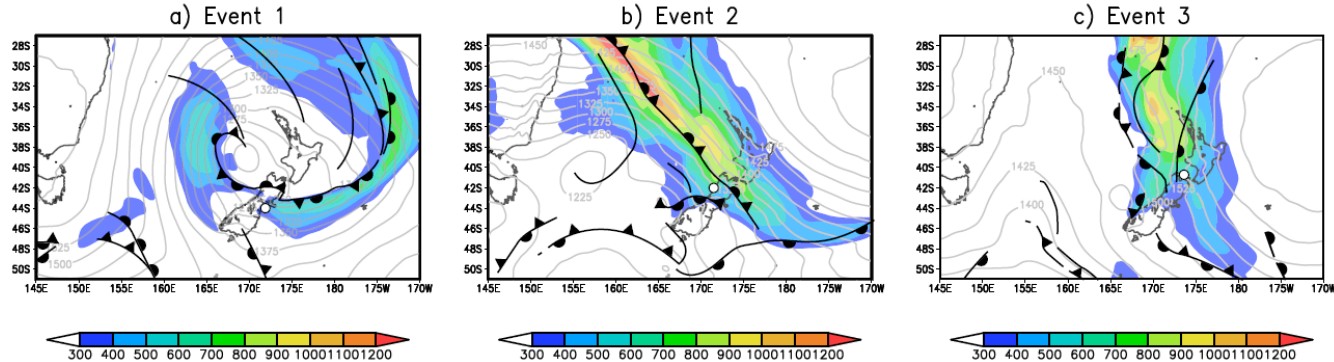

**Figure 1: IVT magnitude (shading) and 850 hPa geopotential (grey isolines), with analysis of frontal activity for each of the three case study events overlain. Event 1: 0000 UTC, 30/05/2021; Event 2: 1200UTC, 16/07/2021; Event 3: 1200UTC, 16/08/2022. The**
**white dots with black outline indicate the approximate location of peak flooding for each event.**

**3.2 Northwesterly Airflow (Event 2)**

Event 2 was associated with a strong northwesterly IVT and airflow (Fig. 1b), with substantially higher IVT in comparison to
event 1. The overall system encompassed zones of exceptional magnitude IVT (>1250 kg m$^{-1}$ s$^{-1}$), corresponding to an occluded
front extending across the Tasman Sea and with the zone of high IVT extending as far away as the Indian Ocean. IVT values
at landfall on the west coast approached 1000 kg m$^{-1}$ s$^{-1}$. Correspondingly, the IVT field matches well with the typical features
of an AR. As well as the relatively high IVT magnitude of this event, a key characteristic was its persistence, pushing it to the
highest category (5) on the Ralph et al. (2019) AR scale and resulting in high precipitation over a three-to-four-day period.

In contrast to event 1, the IVT-EFI forecast at 7-9 days is more spatially coherent at large scales compared to the TP-
EFI and more closely matches the overall pattern of high IVT (Fig. 2, middle row). Despite this stronger large-scale match,
and the higher IVT-EFI across a wider area, the placement of higher values is somewhat to the north of the main area of
extreme precipitation, with the result being that the TP-EFI actually indicates more extreme conditions over the South Island
upper west coast for the time of event 2.

As with event 1, the likelihood of an extreme event is not as obvious from the 10–15-day EFI values (Fig. 3, middle
row), although for event 2 there is a more coherent large-scale pattern in the IVT-EFI values that resembles somewhat both





**Figure 2: 7–9-day EFI forecast for TP, IVT, and IVT minus TP difference for Event 1 (upper row), Event 2 (middle row) and Event 3 (lower row).**





the 7-9 day values and actual IVT situation. As for the 7–9-day IVT-EFI values, the highest values are situated even further northwards of the zone of highest precipitation. Thus, although the 10–15-day TP-EFI values are generally lower and less spatially coherent than for IVT-EFI, they are higher in the regions where extreme precipitation actually occurred.

### 3.3 Northerly Airflow (Event 3)

Event 3 was associated primarily with northerly airflow and IVT, impacting a wide zone across parts of the northern and western North Island, South Island west coast and most of all, the South Island north coast (Fig. 1c). The airmass originated from the tropics and subtropics, and contained a complex series of frontal features with the overall system remaining in-place for approximately four days due to a slow-moving high-pressure system to the east. As with Event 2, zones of exceptional magnitude IVT existed upwind of the coast with landfall values for the South Island north coast that at times exceeded 1000 kg m$^{-1}$ s$^{-1}$ – extremely anomalous values for August from a climatological perspective. This long and narrow zone of high IVT matches closely the typical spatial characteristics of a strong AR.

For the 7–9-day forecast, both the IVT- and TP-EFI show a large zone of very high values (>0.8 – Fig. 2, lower row). These high values – especially at this lead time – are particularly widespread for the IVT-EFI. However, for both the IVT- and TP-EFI (but IVT especially), the highest values are further east compared to the zone of highest precipitation. Unlike events 1 and 2, for event 3 the IVT-EFI values are almost universally higher than for the TP-EFI over the zone of highest precipitation.

Also, and unlike events 1 and 2, at the 10–15-day forecast period the EFI patterns and values clearly indicate a forthcoming extreme event – although the EFI magnitudes are slightly lower than the 7–9-day forecast (in the 0.6-0.8 range – Fig. 3, lower row). However, the spatial dimensions of the high EFI area match well the zone of high IVT (Fig. 1c). The IVT-EFI values are again more extreme than for the TP-EFI, both upwind of and over the area of highest precipitation.

### 4 Discussion and wider implications

Three very different synoptic weather situations and associated extreme weather events have been explored, which represent the main types of AR systems experienced in New Zealand (Prince et al. 2021). Moreover, these three case studies capture the unusual situation in New Zealand of landfalling ARs occurring on coastlines of different orientation with respect to the prevailing atmospheric circulation. Despite the different meteorological processes involved, each case study was associated with significant IVT and precipitation magnitudes that were either close to or record-breaking, resulting in major impacts on human systems.

As with the meteorological situation, key aspects of the EFI for TP and IVT also differ between the events. For the 10–15-day period, indication of a forthcoming extreme event was negligible for event 1 in terms of both TP and IVT, present only for IVT with event 2, but for event 3 was present for both. At the 7–9-day forecast horizon the large-scale EFI patterns approximately matched those of IVT during the events. However, whereas the larger-scale pattern was stronger and more spatially coherent for the TP- (vs. IVT-) EFI in event 1, the IVT-EFI pattern was stronger for events 2 and 3. Finally, in terms




Figure 3: 10–15-day EFI forecast for TP, IVT, and IVT minus TP difference for Event 1 (upper row), Event 2 (middle row) and Event 3 (lower row).





of EFI values specific to the locations of most extreme precipitation, more extreme local conditions were indicated by the TP-EFI for events 1 and 2, but stronger for the IVT-EFI in event 3.

The tendency for the IVT-EFI to be more useful in detecting extreme events at longer forecast horizons in northern hemisphere locations (Lavers et al., 2016, 2017, 2018) is not replicated for all situations in the current case studies. Lavers et al. (2016) noted that for western Europe the IVT-EFI was more useful than the TP-EFI during positive NAO conditions, which are commonly associated with a greater westerly component to the general circulation and more frequent passage of extratropical cyclones. In contrast, in the current study the strongest and weakest IVT-EFI signals were both found during weak westerly flows. It must be noted that in Lavers et al. (2016, 2017, 2018) the EFI was specially calculated for each forecast day, whereas herein we have only employed those currently available in the ECMWF archive. Therefore, the longer averaging periods used for the EFI herein may be in part responsible for the discrepancy in results. Furthermore, event 1 experienced a clear frontal boundary – somewhat at odds with the Lavers et al. (2016) finding that the IVT-EFI was more useful during periods where frontal rainfall dominated. Instead, the much weaker IVT-EFI signal for event 1 may be due to a weaker overall IVT magnitude. Irrespective of the cause, these differences in IVT- versus TP-EFI based forecasts indicate subtle (but important) variation from the identified western European EFI paradigm and suggest further investigation is needed.

The occurrence of the highest IVT-EFI values (and TP-EFI to a lesser extent) further to the north (event 2) or east (event 3) of the most extreme precipitation points to further important features of IVT vs TP forecasts. An equatorward (i.e. northwards, in this case) displacement of the IVT-EFI signal from the core precipitation zone has also been observed in a European context (Cox and Lavers, 2020). However, this phenomenon was thought to be associated with IVT also being influenced by the wider low-pressure system rather than just the frontal boundary where peak precipitation rates typically occur – a context that does not quite fit events 2 and 3 here. In these two cases a full explanation of the offset requires further research. A further matter of interest is that the spatial offset of high IVT-EFI values compared to the actual location of high IVT during events 2 and 3. These offsets correspond to a more downwind location (in the context of broad westerly circulation) of high IVT-EFI values and also to the left/equatorward side of the jet-stream exit. This offset is not apparent for event 1 – under this type of east coast event, New Zealand typically sits between a split jet-stream structure (e.g. Kingston et al., 2022; Prince et al., 2021).

One final matter in the interpretation of the three case studies relates to the match between event duration and the multi-day aggregation of EFI data in ECMWF archive. Duration increased from events 1-3, meaning that while event 3 could occupy the full duration of a 7–9-day forecast aggregation, event 1 could not. This means that depending on the starting day, the 7–9-day forecast for event 1 could include the relatively normal weather conditions before or after this extreme event, as well as the event duration itself – naturally muting the 7–9-day aggregated forecast. This effect was most likely even greater for the 10–15-day EFI, and thus may partly explain the stronger apparent EFI signal detected for the longer duration event 3.

This article is the first investigation to use the EFI in New Zealand and raises at least two possible considerations for future research. First, a more thorough analysis of past extreme events would reveal if these results hold for a larger sample and would allow for a more thorough comparison with earlier results in western Europe. Second, the calculation of the EFI on

daily time steps in week 2 would afford the opportunity of more clearly determining the difference in skill between the IVT and TP EFIs at these longer lead times. Significantly, these results challenge (or at least suggest complexity in) the existing western European paradigm of superior IVT- compared with TP-EFI forecasts under conditions of stronger westerly circulation. Herein, we demonstrate the importance of testing such concepts in different geographical settings, with different

airflows and AR landfall climatologies – this context is important for improving medium-range forecasts and reflecting on underlying forecast model process representation.

**Data availability**

The ERA5 and EFI data used are available through the ECMWF archive (https://www.ecmwf.int/en/forecasts/access-forecasts/access-archive-datasets). MetService analyses are available on request from https://www.metservice.com.

**Author contribution**

Conceptualisation by DGK, DAL and DMH. Data curation by DAL. Formal analysis, investigation and visualisation by DGK and LC. Project administration and supervision by DGK. Writing – original draft preparation by DGK. Writing – review and editing by DGK, DAL and DMH.

**Competing interests**

The authors declare that they have no conflict of interest.

**Acknowledgements**

David Lavers was supported by the Copernicus Climate Change Service which is implemented by ECMWF on behalf of the European Union. This research is a contribution to the UNESCO Chair in Water Sciences at the University of Birmingham, UK.

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
