# Peer review of "Brief communication: Forecasting extreme precipitation from atmospheric rivers in New Zealand"

_EGUsphere, 2024_

## Referee Comment (RC1)

**Brief Communication: Forecasting extreme precipitation from atmospheric rivers in New Zealand**
Kingston et al.,

Proposed for publication in NHESS

Round 1 · July 2024

**Reviewer Report**

The manuscript, entitled "Brief communication: Forecasting extreme precipitation from atmospheric rivers in New Zealand," examines the forecasting of extreme precipitation from atmospheric rivers in New Zealand. It focuses on the application of the European Centre for Medium-Range Weather Forecasts (ECMWF) Extreme Forecast Index (EFI) for two variables: total precipitation (TP-EFI) and vertically integrated water vapor transport (IVT-EFI). The results indicate that TP-EFI may offer a more accurate representation of extreme precipitation events than IVT-EFI, which is contrary to previous research in Europe. The study analyzes three recent significant extreme precipitation events in New Zealand, evaluating the effectiveness of the EFI in improving medium-term forecasting of these events.

The manuscript addresses an intriguing and, as it states, novel question. The three case studies are thoroughly analyzed, and the databases utilized are of high quality. Furthermore, the topic has a high social impact and the manuscript is well written. However, there are a number of relevant issues that prevent me from approving the manuscript for publication in its current form. These and other issues are detailed in the following paragraphs.

**Major Comments**

1.- In my view, the most questionable aspect of the article is that it is based on the analysis of only three case studies. I understand that the article is intended as a brief communication, but the authors provide conclusions that seek to analyze the general behavior of the tools evaluated in the region of interest. I believe that with only three case studies, it is not possible to obtain generalized conclusions. Furthermore, the title is overly ambitious for the analysis that has been carried out. I do not expect the authors to perform a climatologically based analysis (although it would be a valuable addition to the study and I do not see any impediment for them to do so). However, if they persist in their idea of using only three case studies, this should be incorporated in the title and it should be made clear that it is a preliminary analysis that should be completed with a much larger database of cases.

2.- Additionally, the authors fail to address another pertinent question: how do they account for the discrepancies observed in previous European studies? It is acknowledged that the authors currently lack an explanation for this phenomenon. However, incorporating a number of potential hypotheses that could elucidate this behavior would be a valuable contribution.

**Minor Comments**

**L43,44.-** Please confirm whether the result regarding prediction utility as a function of NAO phase is statistically significant.

**L112.-** Do the authors consider the scaling thresholds used by Ralph to categorize AR events to be appropriate for New Zealand? Other authors have had to adapt the thresholds of that scale, as it was shown that they were not the most appropriate for a region other than the American West Coast (see, for example, *European West Coast atmospheric rivers: A scale to characterize strength and impacts*). I am not saying that it is not, I am saying that it might be necessary to determine, based on the climatology, whether the thresholds are the most appropriate for the region.

**Figure 1:** Just out of curiosity, how have the authors made the front maps?

.

---

## Author Response (AR1)

**Brief communication: Forecasting extreme precipitation from atmospheric rivers in New Zealand**

**Response to RC1**

Reviewer Report

The manuscript, entitled "Brief communication: Forecasting extreme precipitation from atmospheric rivers in New Zealand," examines the forecasting of extreme precipitation from atmospheric rivers in New Zealand. It focuses on the application of the European Centre for Medium-Range Weather Forecasts (ECMWF) Extreme Forecast Index (EFI) for two variables: total precipitation (TP-EFI) and vertically integrated water vapor transport (IVT-EFI). The results indicate that TP-EFI may offer a more accurate representation of extreme precipitation events than IVT-EFI, which is contrary to previous research in Europe. The study analyzes three recent significant extreme precipitation events in New Zealand, evaluating the effectiveness of the EFI in improving medium-term forecasting of these events.

The manuscript addresses an intriguing and, as it states, novel question. The three case studies are thoroughly analyzed, and the databases utilized are of high quality. Furthermore, the topic has a high social impact and the manuscript is well written. However, there are a number of relevant issues that prevent me from approving the manuscript for publication in its current form. These and other issues are detailed in the following paragraphs.

Major Comments

1.- In my view, the most questionable aspect of the article is that it is based on the analysis of only three case studies. I understand that the article is intended as a brief communication, but the authors provide conclusions that seek to analyze the general behavior of the tools evaluated in the region of interest. I believe that with only three case studies, it is not possible to obtain generalized conclusions. Furthermore, the title is overly ambitious for the analysis that has been carried out. I do not expect the authors to perform a climatologically based analysis (although it would be a valuable addition to the study and I do not see any impediment for them to do so). However, if they persist in their idea of using only three case studies, this should be incorporated in the title and it should be made clear that it is a preliminary analysis that should be completed with a much larger database of cases.

- *Author response:* Thank you for these comments. We are pleased the Reviewer acknowledges our submission as a brief communication that highlights exciting new findings; and we intended this paper format to be a call for the community to explore further. In this context, we have re-written the final (concluding) paragraph in the manuscript and have developed a new, more specific title: "*Forecasting extreme precipitation from atmospheric rivers in New Zealand – case studies under different synoptic situations*".
  We acknowledge that a broader analysis would be valuable, but this study was only intended to provide a first look at the applicability of previous European-focussed findings in a different climatological situation where ARs play a major role in the occurrence of extreme hydroclimatic events. Nonetheless, we believe the results presented in this brief communication are interesting in their own right, while also indicate the need for further research, not just in New Zealand but also other physical

settings where extreme IVT values (such as ARs) are important for extreme precipitation events.

2.- Additionally, the authors fail to address another pertinent question: how do they account for the discrepancies observed in previous European studies? It is acknowledged that the authors currently lack an explanation for this phenomenon. However, incorporating a number of potential hypotheses that could elucidate this behavior would be a valuable contribution.

- *Author response:* Accounting and explaining these discrepancies is an important point, and we have strengthened the discussion is this respect. However, we would like to emphasise that directions for further research were already noted in the original manuscript: obtaining EFI output on daily scales (as noted in the existing manuscript on lines 197-198), and secondly, an expanded analysis that moved beyond the case study approach (lines 196-197).
  In terms of further hypotheses that could be tested, we now highlight additional features of the results that warrant further investigation, specifically linked to duration (line 173). The longest event (3) had the strongest IVT signal, whereas the shortest event (1) had the strongest TP signal. The AR associated with event 3 was also slower moving than event 2 and especially event 1. The magnitude of orographic- vs. synoptic-generated precipitation may also be important – so the magnitude of frontal uplift compared to the angle at which the AR interacts with topographic features (and the magnitude of those features) would be a further matter for additional study.

Minor Comments

L43,44.- Please confirm whether the result regarding prediction utility as a function of NAO phase is statistically significant.

- *Author response:* Thank you for the question. To investigate the statistical significance of differences in the Relative Operating Characteristic (ROC) areas between the water vapour flux (IVT) and precipitation on NAO-positive days, Lavers et al. (2016) used a bootstrap procedure. Results for forecast day 9 in Figure 3 in Lavers et al. (2016) show that the interquartile ranges of the bootstrapped distributions of the IVT and precipitation do not overlap, suggesting that there is statistical evidence for a difference in ROC areas and that the IVT EFI is more useful than the precipitation EFI during the positive NAO phase. We do note, however, that significance at the 90 or 95% levels was not found. Conversely, the IVT is less useful than the precipitation during the negative NAO phase. Brief summary of these additional details are now provided in the manuscript (line 43).

L112.- Do the authors consider the scaling thresholds used by Ralph to categorize AR events to be appropriate for New Zealand? Other authors have had to adapt the thresholds of that scale, as it was shown that they were not the most appropriate for a region other than the American West Coast (see, for example, European West Coast atmospheric rivers: A scale to characterize strength and impacts). I am not saying that it is not, I am saying that it might be necessary to determine, based on the climatology, whether the thresholds are the most appropriate for the region.

- *Author response:* In the New Zealand context, Prince et al. (2021) have previously used the Ralph et al. thresholds. With some exceptions, the thresholds were shown to perform consistently in terms of matching increasing level of precipitation for higher ranked events, but without formal analysis of whether the switch from primarily beneficial to harmful impacts occurred at the same point. We  now note the previous use of these thresholds in NZ (line 71).

Figure 1: Just out of curiosity, how have the authors made the front maps?

- *Author response:* ERA5 IVT and 850 hPa geopotential were plotted using GrADS. The type and location of frontal boundaries were determined by the Meteorological Service of New Zealand from their operational analyses – with the two maps subsequently overlain here to produce Figure 1.

**Response to RC2**

The manuscript titled: Brief Communication: Forecasting extreme precipitation from atmospheric rivers in New Zealand compares the Extreme Forecasting Index (EFI) for IVT and precipitation on medium-range timescales for three case studies of extreme precipitation over New Zealand. They find the variable with the better forecasting capability depends on the synoptic set up. The topic is important for improving forecasts of natural hazards in New Zealand and the manuscript is well written. Given there isn't enough data to make general conclusions (understandable for a brief communication) and the comparison with the European studies aren't that useful due to the major methodological differences (case studies vs larger sample size, daily vs multiday EFI), the manuscript could be reframed to emphasise the current gaps and avenues for improving extreme precipitation forecasts in New Zealand using EFI as an example, as I think this will enhance the value of the article. I recommend major revisions.

- *Author response:* Again, we are pleased the Reviewer acknowledges our submission as a brief communication and notes the importance of the topic and this case study contribution, which open avenues for further research.
- We have re-written the concluding paragraph of the manuscript according to this suggestion, and made a corresponding change in emphasis in the final sentence of the abstract.

Major Comments:

The observations in Figure 1 are for one timestep whereas the forecasts are multiday averages. It would be fairer to compare the forecasts with multiday averages of the observations. This may alter the conclusions and may explain some of the spatial offsets observed between the observations and forecasts.

- *Author response:* The reviewer raises an interesting possibility here. Following this suggestion we have investigated multiday averages of IVT corresponding to the EFI aggregation periods; but this provides no resolution or explanation to the spatial offsets observed for the IVT-EFI. For event 1 the multiday mean IVT pattern is less extreme but with similar spatial co-ordinates. For event 2 the multiday mean zone of high IVT is more spatially dispersed but does not align with the spatial offset with the IVT-EFI. For event 3 there is little difference in location or magnitude between the instantaneous and 3-day average plots. We summarise this comparison briefly in the revised manuscript (line 178).

The Lavers work showed higher predictability from IVT over a season (presumably there were cases within that season where the precipitation EFI yielded higher predictability). So, I think the authors should be careful comparing their results given the difference in sample size. i.e. the difference between the NZ and European results may not be geographical. You could focus the discussion more on the future work that could be done for NZ with Europe as an example rather than as a comparison.

- *Author response:* Thank-you for this comment. Following the similar comment from Reviewer 1 we intend to edit our conclusions to be mindful of the more limited scope of this initial investigation compared to the Lavers et al. European work.

Minor Comments:

The methods needs more information on the forecast system – what model, model version, resolution etc.

- *Author response:* Thank you for the comment and we apologise for leaving this information out of the original article. The EFI is calculated using the ensemble forecasts from the ECMWF Integrated Forecasting System (IFS). For the events, two different versions of the IFS were used, both at O640 (~18km) resolution: IFS Cycle 47r2 for Events 1 and 2; and IFS Cycle 47r3 for Event 3; further details of the IFS are available at https://www.ecmwf.int/en/forecasts/documentation-and-support/changes-ecmwf-model. See lines 74-75.

Figures 2 & 3: It would be helpful to indicate on the plots where the peak precipitation occurred or overlay precipitation contours.

- *Author response:* the locations of peak precipitation are already indicated on Figure 1.

Events 2&3 were forecast during a positive SAM. Since you discuss the NAO in the NH context, it may be worth discussing the SAM too.

- *Author response:* Thanks for the suggestion to consider the phase of SAM. However, based on the daily AAO index from the NOAA CPC (https://www.cpc.ncep.noaa.gov/products/precip/CWlink/daily_ao_index/aao/aao.shtml) there is little evidence of positive SAM situations during the three case studies. Specifically:
  Event 1: On day of forecast, SAM=-0.54. Then -2.3 on the day before the event, -1.6 for the first day of the event
  Event 2: On day of forecast, SAM=-1.50. Then -0.98 on day before event, -1.0 for the first day of the event
  Event 3: On day of forecast, SAM=0.00. Then -0.59 on day before event, -1.06 for the first day of the event, moving to neutral then 1.25 on the fourth day, then 1.68 on the fifth day.

  We will make note of this in the revised manuscript and make this further point of comparison against the previous Lavers et al. work (lines 89, 111, 134, 167).